# Breast Imaging Physics in Mammography (Part II)

**DOI:** 10.3390/diagnostics13233582

**Published:** 2023-12-01

**Authors:** Noemi Fico, Graziella Di Grezia, Vincenzo Cuccurullo, Antonio Alessandro Helliot Salvia, Aniello Iacomino, Antonella Sciarra, Daniele La Forgia, Gianluca Gatta

**Affiliations:** 1Department of Physics “Ettore Pancini”, Università di Napoli Federico II, 80127 Naples, Italy; 2Radiology Division, ASL Avellino, 83040 Avellino, Italy; graziella.digrezia@gmail.com; 3Department of Precision Medicine, Università della Campania “Luigi Vanvitelli”, 80013 Naples, Italy; vincenzo.cuccurullo@unicampania.it (V.C.); antoniosalvia89@gmail.com (A.A.H.S.); gianluca.gatta@unicampania.it (G.G.); 4Department of Human Science, Guglielmo Marconi University, 00193 Rome, Italy; nelloiacomino@libero.it; 5Department of Experimental Medicine, Università della Campania “Luigi Vanvitelli”, 80013 Naples, Italy; antonella.sciarra@unicampania.it; 6IRCCS Istituto Tumori “Giovanni Paolo II”, 70100 Bari, Italy; d.laforgia@oncologico.bari.it

**Keywords:** breast imaging, medical physics, mammography

## Abstract

One of the most frequently detected neoplasms in women in Italy is breast cancer, for which high-sensitivity diagnostic techniques are essential for early diagnosis in order to minimize mortality rates. As addressed in Part I of this work, we have seen how conditions such as high glandular density or limitations related to mammographic sensitivity have driven the optimization of technology and the use of increasingly advanced and specific diagnostic methodologies. While the first part focused on analyzing the use of a mammography machine from a physical and dosimetric perspective, in this paper, we will examine other techniques commonly used in breast imaging: contrast-enhanced mammography, digital breast tomosynthesis, radio imaging, and include some notes on image processing. We will also explore the differences between these various techniques to provide a comprehensive overview of breast lesion detection techniques. We will examine the strengths and weaknesses of different diagnostic modalities and observe how, with the implementation of improvements over time, increasingly effective diagnoses can be achieved.

## 1. Introduction

In Italy, breast cancer is the most frequently diagnosed neoplasm [1], and thanks to screening, mortality rates are lowered [2] since prevention, like early diagnosis, allows for more effective treatment [3]. With the increase in the number of screenings, there has been an increase in the incidence of breast cancer, probably due to more extensive diagnostic investigations [4,5,6,7].

Mammography is the gold standard for breast cancer diagnosis [8], as it is low cost, low administered radiation dose [9], and high sensitivity [10]. However, as we have seen in previous work [11], there are also critical issues that are sometimes overcome [12], either by the use of higher performance machines or by more in-depth investigation techniques [13].

In order to make more effective diagnoses, quality images are required, and in difficult cases such as high densities, a more in-depth examination may be necessary [14,15,16,17,18,19].

In particular, we have seen how the machinery works to generate the X-ray beam, and we have observed how the characterization of the X-ray tube is crucial for obtaining high-quality diagnostic images [11].

Machine settings are extremely important, both in terms of the quality of the obtained image and in dosimetric terms, to provide the patient with the lowest possible dose. Since the breast is irradiated, other sensitive organs are also exposed to the beam. Tightening the beam and setting the tube appropriately contribute not only to obtaining quality images, but also to delivering lower doses. Having examined the mammographic technique in detail, this paper considers the main techniques that can be combined with or sometimes replace mammographic examination.

We will explore techniques such as Contrast-Enhanced Mammography (CEM) or Digital Breast Tomosynthesis (DBT), observing their differences both from a technical standpoint and in terms of diagnostic objectives. We will also observe the substantial differences between these various techniques, allowing us to have a clear understanding of when to prefer one over another.

We will then delve into functional imaging in nuclear medicine and touch on image processing, aiming to provide a comprehensive overview of major diagnostic methodologies in the field of breast imaging.

In the following section, we will look at specific techniques for breast cancer diagnosis and observe the strengths and weaknesses of different diagnostic modalities [20,21,22,23,24,25].

## 2. Breast Tomosynthesis

Breast Digital Tomosynthesis is a technique used both as a screening tool for early breast cancer detection and a diagnostic tool for evaluating anomalies detected during clinical examination or mammography [26].

Breast Digital Tomosynthesis (DBT) is achieved using a tube that generates a beam of photons, with information related to the differential attenuation of X-rays as they pass through various structures that make up the patient’s area of interest [27].

The patient is in an anatomical position (standing with hands against the tube), parallel to the tube, and the breast is perpendicular to the generated beam. The breast is compressed to ensure even distribution and to prevent motion during image acquisition, which could generate artifacts [28,29,30].

The tube acquires an initial static image of the breast to obtain an initial overview and provide an understanding of the breast’s characteristics. Subsequently, images at discrete angles are acquired by rotating around the breast for a total of N. The acquisition is not along the circumference but at discrete angles, enhancing the ability to detect details and hidden structures [31,32].

Individual two-dimensional (2D) sections are reconstructed, resulting in a pseudo-three-dimensional image that provides a detailed view of breast structures. The acquired images are then reconstructed, pre-processed, and the DBT is finally obtained [33,34].

The patient’s position varies with each acquisition, and Breast Digital Tomosynthesis can be performed in various modes, providing different views. In the Mediolateral Oblique (MLO) projection, the patient stands with her raised arm while the tube rotates, while in the Cranio-Caudal (CC) projection, the tube is at a 90-degree angle, and the patient is standing [35,36,37].

The radiologist reviews the images, analyzing the different sections, focusing on the suspicious area and selecting a specific Region of Interest (ROI). [38] The suspicious area typically shows significant attenuation and may not be present in all sections. It may have irregular margins, microcalcifications, and calcium formations typically found within the lobular ducts [39]. Of course, there are also macrocalcifications, which are larger but not visible in these sections [40,41,42].

The initial diagnosis is made visually by recognizing suspicious structures and localizing the area for biopsy. For the biopsy, the patient is punctured in the suspicious area, and a sample of cells is collected through needle suction [43,44,45,46]. The histology by a pathologist is awaited, who studies the nature of the cells, either individually or in groups [47].

There are different classifications of tumors based on location (always within the mammary gland), such as lobular, intraductal, or pinwheel structures (distortions). [43,44,45,46,47,48,49,50]

New studies are implementing a machine learning algorithm (deep learning) by which Regions of Interest (ROI) obtained from the images can be used to verify the presence or absence of pathology [51,52,53,54]. The network identifies an unhealthy patient from a healthy one, attempting to locate the lesion [55]. This software is known as CAD (Computed Added Detection) [56,57,58].

There are differences between digital mammography (FFDM) and Breast Digital Tomosynthesis [59].

Mammography provides a two-dimensional image in which overlapping structures may hide anomalies, especially in cases of high breast density, potentially leading to false positives or false negatives [60]. In Breast Digital Tomosynthesis, the effect of tissue overlap is significantly reduced, resulting in increased sensitivity for detecting abnormalities and providing a more comprehensive and detailed view of breast structures [61,62].

Regarding dosimetry, mammography, being a single two-dimensional projection, involves a lower dose of radiation compared to Breast Digital Tomosynthesis [63,64]. The cumulative radiation dose for obtaining 3D images is slightly higher in Breast Digital Tomosynthesis, although it remains low [65,66,67,68].

Another difference is the time required for the examination. Traditional mammography is a faster and simpler examination, whereas Breast Digital Tomosynthesis involves greater data processing and longer examination times [69,70,71,72].

In terms of costs, traditional mammography is generally less expensive compared to Breast Digital Tomosynthesis, which requires specialized hardware and software for the acquisition and interpretation of 3D images [73,74,75].

In summary, Breast Digital Tomosynthesis is advantageous over traditional mammography due to its ability to provide more detailed information, reduce tissue overlap effects, and improve diagnostic capabilities while reducing false positives and negatives. However, the choice between the two often depends on availability, cost considerations, and the specific needs of the patient [76,77,78,79].

## 3. Radionuclide Imaging

Nuclear Medicine is based on the concentration of radioisotopes in living cells and tissues [80].

Nuclear Medicine has unique characteristics in diagnostic imaging because, unlike other methods, the image is based on differences in concentration and not density. [81] The examination is only possible on living humans or animals and expresses pathophysiological and molecular assumptions that provide original information compared to those with a more strictly morphostructural imprint [82].

If it is true that biological premises are also the basis of Magnetic Resonance Imaging and the use of radiological contrast agents, it is only in Nuclear Medicine that imaging is, by definition, “functional”. In fact, the possible marking with radioisotopes of the most important molecules of biological interest, cells, and drugs allows us to trace the pathophysiology of all the functions of the human body, obtaining information that can allow for early diagnosis and a better connection with prognosis and therapy. The concept of the tracer expresses the nodal point of radioisotopic methods, much more than the nuclear radiation that gave its name to the discipline. In fact, if it is the radionuclide that allows for the external visualization of a radiocompound injected into a patient, it is the tracer that designs its behavior and capabilities, creating the conditions for a concentration that becomes the tool for understanding the molecular alteration and/or or pathophysiology that characterizes the disease [83].

In recent years, Nuclear Medicine imaging has assumed an important role in study and research, mainly in cardiology [84], neurology [85], and, above all, in oncology [86], due to its ability to produce high-resolution three-dimensional images of alterations in the metabolism of the human body, i.e., biochemical processes and cell activity, even before structural changes are highlighted [87,88,89,90]. These images are formed with computed tomographic methods that reconstruct the distribution in space of radiopharmaceuticals labeled with short half-life radionuclides and with reduced radiotoxicity, specifically injected into the patient [91]. The concentration of a radiocompound is present exclusively in the presence of ”functionally active” cells at the level of the territory examined, not being possible, for example, where there is fibrosis or necrosis [92]. Concentration variations and, therefore, the definition of a pathological event can precede the variations of the morpho-structural characteristics of a lesion [93]. Diagnostic imaging with radionuclides is an important method of investigation for many oncological pathologies. It provides both the evaluation of physiological and anatomical data, and information to easily guide the study, treatment, and follow-up of patients suffering from breast pathologies [94]. Scintimammography has also been shown to provide additional data that may change the planning strategy for primary breast cancer surgery in up to 10% of cases. Scintimammography provides additional information about the true extent of the disease so that surgical planning can be more accurately undertaken with the aim of complete surgical removal of breast cancer in a single operative procedure [95]. Nuclear Medicine also offers other very valid approaches, such as PET and Radioguided Surgery, especially useful in staging, therapy, and follow up of breast cancer [96,97,98].

## 4. CEM

Mammography, is the most widely used technique for symptomatic patients over 40 years of age and also for breast cancer screening of the entire population [99]. The technique is readily available and inexpensive, but it lacks sensitivity, especially for young patients, reflecting the increased breast density in this patient group. In the last two decades, spectral contrast-enhanced mammography (CEM) has been introduced. It is a technique that combines intravenous administration of iodinated contrast medium and digital mammography, exploiting the dual-energy technique to improve lesion detection [100]. It is also based on dual-energy 26–33 kVp and 44–50 kVp breast exposure after contrast administration, which makes pre-contrast exposure unnecessary. It exploits the neovascularisation of breast tumors in a similar way to breast MRI, providing a more precise imaging element. This is useful for subsequent action, which may be primary surgery or neoadjuvant chemotherapy. While the gold standard for breast cancer diagnostic investigation is MRI, the latter is costly, time-consuming, and prone to false positives. CEM may be an effective alternative due to the combined action of an iodinated contrast and conventional mammography technique, providing greater diagnostic accuracy, especially in patients with denser parenchymal background patterns, and may increase access to vascular imaging while reducing examination costs [101].

CEM allows the visualization of improved findings compared to normal unenhanced breast tissue by exploiting the increased contrast uptake of malignant neoplasms, becoming useful in the diagnosis and staging of primary breast cancer [102].

Retrospective reading studies have shown a significant improvement in the sensitivity and specificity of CEM to detect breast carcinomas compared to standard two-dimensional (2D) mammography. The improvement in sensitivity is due to the ability of CEM to identify tumors that would normally be masked by denser breast parenchyma on conventional mammography. Improvements in sensitivity with CEM are observed in women with denser background patterns and in younger pre-menopausal women, where a >20% improvement in sensitivity is possible. The equipment is similar to the mammogram, but a titanium or copper filter is added to obtain the high-energy image, followed by the use of post-processing software. CEM is performed with a single breast compression following injection of iodinated contrast agent (1.5 mL/kg body weight). MLO and CC projections are then performed on each breast, two minutes after the injection, followed by a double exposure of high and low energy. The low-energy exposure uses the same X-ray energy spectrum as a standard full-field digital mammogram (FFDM) with a kilovoltage peak of about 30 KVp, depending on the thickness and density of the compressed breast. The low-energy component is acquired using an X-ray spectrum identical to an FFDM, so it is not surprising that the image resembles a conventional mammogram. The high-energy exposure exploits the K-edge of the iodine, having a peak kilovoltage of about 45 KVp. As this technique has a double exposure of high and low energy, the radiation dose is therefore higher than FFDM. Using the dual-energy weighted logarithmic subtraction technique, images are produced, although the mode is less sensitive to motion artifacts than traditional temporal subtraction, however, with CEM, the times are more dilated [103].

The recombined image is also subject to a ’rim’ artifact with increased density at the periphery of the breast due to radiation scattering; an artifact that can be reduced by special scatter correction techniques. For each acquisition, lasting from 2 s to 20 s, depending on breast thickness and machine, we obtain high- (44–50 kVp) and low-energy (26–33 kVp) images. Both diagnostic images contribute to anatomical morphological information and the detection of possible lesions. High-energy imaging is obtained using energies above the k-edge of Iodine, which is not useful for diagnostic purposes but is useful for recombination imaging. The images are post-processed with dedicated software, and by using high- and low-energy images, a recombined image is obtained, minimizing the overlap of glandular tissue by maximizing the spread of contrast in the irradiated area. There are no standardized methodologies for the acquisition of contrast medium, dose, flow rate, contrast medium administration interval, and image acquisition, but reference is made to what was published in 2009 by C. Dromain, C. Balleyguier, G. Adler, J. R. Garbay, S. Delaloge, Contrast-enhanced digital mammography [104].

For the administration of contrast medium, the guidelines and precautions used in CT can be considered. In particular, the contrast medium is injected manually or preferably via an automatic injection pump at a rate of 2–3 mL/s, followed by a bolus of saline solution. After injection, the visualization time useful for image acquisition is about 10 min after administration. In particular, the acquisition can take place 2 to 2.5 min after administration within 10 min (including injection time), which is the final time for image acquisition.

The dual-energy technique involves double exposure for each projection, so the radiation dose is higher than in standard mammography. For low-energy exposure, we have a radiant dose superimposed on that of a digital mammogram, while for high-energy exposure, we have an overall increase in radiant dose of between 20% and 80%, depending on the type of equipment used, the system setting, and the thickness of the breast.

This remains below the threshold of the European guidelines for screening mammography and the Mammography Quality Standards Act guidelines that state that increased dose is not a significant lifetime risk factor. CEM has high accuracy both in measuring the main lesion and in identifying multifocality and multicentricity of lesions, representing a 5–46% increase in sensitivity and a 3–15% increase in specificity over standard mammography. The limitation in comparison with MRI is the impossibility of an adequate study of the axillary cavities due to the overlapping field of view with mammography [105].

The administration of an iodinated contrast medium is essential for the examination. It carries a small risk of allergic reactions (<1% of cases), and they are usually mild and self-limiting. Another potential problem with iodinated contrast agent is contrast agent-induced nephropathy, so protocols have been developed to reduce the risk of acute renal damage, suggesting an assessment of renal function prior to contrast agent administration. Although CEM significantly improves the accuracy of local staging, false positives and false negatives still occur. A recognized reason for a false negative is when the lesion is not included in the mammographic field of view, but CEM compares favorably with MRI for local staging of breast cancer. In several studies of women with known carcinomas, CEM approaches the sensitivity of MRI with superior specificity.

## 5. Notes on Processing

Mammography requires specific characteristics related to the type of instrumentation used and the exposure technique, with geometric parameters and beam energy. Of considerable importance for diagnostics are the characteristics of the detector and intrinsic processing. Processing, in particular, has the primary objective of limiting the effect of the thickness gradient between the part of the organ closest to the chest wall and the more distal part, which will have a decreasing thickness and attenuation [106]. In Figure 1, we provide examples of processing. The raw image (a) has low intrinsic contrast, and (b) varies the windowing by selecting a LUT suitable for more superficial regions, thus enhancing the visualization of peripheral regions with lesser thickness and reduced contrast on the internal tissue at the same time. [107] Also in mammography, as in traditional radiology, it is possible to find the application of LUTs after automatic segmentation of the grey level histogram, relative to tissues of interest: non-compressed adipose tissue, compressed adipose tissue, glandular tissue, and pectoral muscles. Other algorithms exploit unsharp masking, CLAHE histogram equalization, or frequency space analysis. In (c), the contrast was optimized by choosing a LUT suitable for more internal regions, but with a loss of information relative to the more superficial regions. Processing was performed by means of equalization-based processing algorithms, histogram in (d), and peripheral in (e), respectively. The latter is done by first applying a low-pass filter, typically Butterworth, with a cutoff frequency of 0.05 cycles per mm, from which an attenuation map is derived and compared to the original image to even out the thickness-related luminance gradient and allow for a uniform representation of the glandular structure [108].

## 6. Conclusions

As we have seen, both in this paper and the first part, breast diagnostics begin with a physical examination that encompasses different techniques, and it is useful to understand the various contexts of use and diverse final goals of different examinations. It is important to specifically understand first-level investigations that use traditional mammographic techniques employed for screenings, especially in Italy [109,110,111].

This can be complemented by other techniques such as digital tomosynthesis (DBT) and Contrast-Enhanced Mammography (CEM), whose basic physical principles, operation, usage context, and diagnostic objectives we have discussed. Starting from mammography, we observed how the 2D diagnostic image can be misleading in visualizing certain details, as two-dimensionality can easily result in the loss of information on less visible or parenchyma-hidden lesions, especially in cases of high density.

The standard technology of mammography can be improved in various ways: by using appropriate machine settings, leveraging physical knowledge to achieve high-quality images with minimal noise to better distinguish tissues and lesions, or by implementing contrast techniques. These techniques, by exploiting tumor vascularization, can detect even small tumor masses and differentiate them from microcalcifications.

To obtain more specific exams, new supporting techniques or exams have been introduced over the years, such as breast digital tomography providing 3D reconstruction, and Contrast-Enhanced Mammography, which, through the injection of iodinated contrast, shows the angiogenesis of tumor masses, as discussed in this paper.

We also briefly touched on nuclear medicine and the use of radionuclides in breast cancer. These techniques differ from mammographic or tomographic analyses, as they involve the use of radioisotopes. Knowing that nuclear medicine is based on the concentration of radioisotopes in living cells and tissues, it possesses unique characteristics in imaging. In nuclear medicine, the image is, by definition, “functional,” utilizing the labeling with radioisotopes to trace the pathophysiology of all functions of the human body, obtaining information for early diagnosis and better connection with prognosis and therapy. Three-dimensional images can also be obtained here using tomographic methods that reconstruct a distribution map of the radioisotope in space, providing useful diagnostic information.

Lastly, we discussed some aspects of image processing with examples. We observed that, for diagnostics, not only is the role of the detector characteristic fundamental but also intrinsic image processing. The choice of Look Up Table (LUT) and Region of Interest (ROI) is crucial, as they provide the radiologist with optimized visualization according to specific needs.

Through the use of algorithms, image corrections are made, with specific filters, to improve image visualization. Image processing importance is not only useful in the assessment phase for the radiologist but also in the objective evaluation phase performed through artificial intelligence. In fact, in recent years, thanks to advancements in electronic devices’ sensitivity and precision, coupled with neural network implementations, remarkable results have been achieved in imaging, providing substantial support in diagnostics. The concept of radiomics has emerged, applicable to most breast imaging modalities such as tomosynthesis, magnetic resonance, or ultrasound. It aims to make image reading as objective as possible and contribute to the radiologist’s visual analysis. This is extremely useful in breast imaging, especially as an objective support for the radiologist’s diagnosis.

There are objective limits in image reading that can be overcome with artificial intelligence analysis. Through this, we can identify the presence of a tumor, qualitatively define its precise anatomical location, morphological characteristics, margins, and extension into surrounding structures, surpassing visual analysis accuracy.

The method involves acquiring an image in a standardized way using suitable reconstruction algorithms, segmenting ROIs or volumes of interest automatically or semi-automatically, extracting various types of features within an ROI of a radiographic image (mammographic in the case of breast cancer). These features can include volume, shape, surface, density, intensity, position, and relationships with surrounding tissues.

Here again, especially regarding mammography, the importance of obtaining correctly acquired, processed, and filtered images is highlighted, as discussed in the paper. Furthermore, acquiring radiomic data combined with data from genetic analyses can provide additional information. Therefore, studying the correlation between image-derived data and tumor molecular characteristics or obtaining new information on tumor phenotype and the microenvironment is possible. The goal is to create prediction models.

The field that synthesizes information related to radiomic images and genetics is called radiogenomics. The combination of these different pieces of information obtained from different data types allows us to obtain information about tumor characteristics and subtypes, directing the most appropriate therapeutic choice.

In summary, knowledge of physical theory and instrument characterization allows us to obtain high-quality diagnostic images and information. This information can be subjected to radiologists in conjunction with artificial intelligence analysis. The higher the quality of image acquisition and processing, the more effective the diagnosis will be. Additionally, it is fundamentally important to choose the right examination to pursue the right goal optimally. We have seen how different techniques can better adapt to specific circumstances, considering visualizations, costs, and dosimetry.

Sophisticated tools and accurate analysis through artificial intelligence can only achieve effective, fast, and precise personalized diagnoses through proper use and with a comprehensive understanding of tool functionality at the right time. In the future, the combination of techniques to obtain diagnostic images combined with the use of artificial intelligence may be useful in providing highly personalized medicine.

## Figures and Tables

**Figure 1 diagnostics-13-03582-f001:**
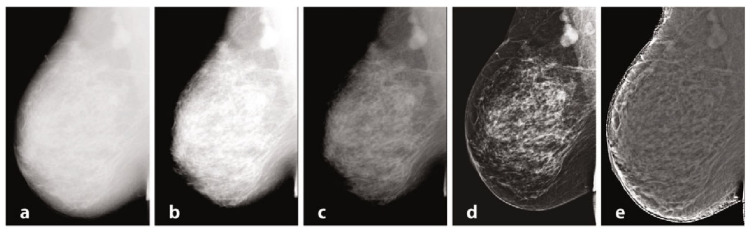
Mammography, different elaborations.

## Data Availability

Not applicable.

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
