# Peer review of "Breast Imaging Physics in Mammography (Part II)"

_diagnostics, 2023, doi:10.3390/diagnostics13233582_

Round 1

Reviewer 1 Report (New Reviewer)

Comments and Suggestions for Authors

The abstract needs quantification. Even though the article resembles a review but not expressed the core values of the review. All the 48 references are cited in the group manner not as a single citation. Design of methodology, identification of research gap and possible solution to this are missing in the paper. From the figures the manuscript may be looks like a case study while going through the article we understand that it is not a case study. The discussion section may be added. The conclusion may be modified. The paper should have a clear distinction of a review or case study one.

Comments on the Quality of English Language

NIL

Author Response

The language throughout the entire article has been revised, sources have been improved, and a significant portion of the article has been reworked.

The abstract and conclusions have been enhanced.

This article constitutes the second part of a previously published article with the same structure, where a physical analysis of mammographic technique was addressed.

This serves as a continuation where, from a physical standpoint, the remaining techniques commonly used in breast imaging are further explored. Comparisons between the techniques have been added to provide a more comprehensive overview, both from a physical and diagnostic objective standpoint.

In both the first and second parts, there is no specific clinical case analyzed, but, as specified, there are examples, such as image processing types. The image found in the last section is, in fact, an example aimed at demonstrating various processing techniques, not for a specific case but to highlight the most important aspect, which is the processing technique.

Reviewer 2 Report (New Reviewer)

Comments and Suggestions for Authors

The review mainly depends on another 

review published before

This should not be the case, this review 

should be independent and citation for the

previous review article should be mentioned 

Also a summarizing tables and figures should

be added to the manuscript 

The manuscript also needs more linguistic and grammar revisions

Comments on the Quality of English Language

The manuscript has to be revised in terms of 

linguistic and grammar edits

Author Response

The entire article has been language revised, sources have been improved, and a significant portion of the article has been reworked, with reference to the first part of the article. The abstract and conclusions have been enhanced.

This article constitutes the second part of a previously published article with the same structure, wherein a physical analysis of mammographic technique is undertaken. This serves as a continuation where the remaining techniques commonly used in breast imaging are explored from a physical standpoint. Comparisons between techniques have been added to provide a more comprehensive overview, both in terms of the physical aspects and diagnostic objectives.

In both the first and second parts, there is no analyzed clinical case; hence, we have not included tables with numerical values. However, we have taken the suggestion to incorporate more specific comparisons between the various techniques to provide a more thorough and insightful overview.

Reviewer 3 Report (New Reviewer)

Comments and Suggestions for Authors

This MS reviews some high-sensitivity diagnostic techniques, which are crucial for early BC diagnosis, such as contrast-enhanced mammography (CEM), digital breast tomosynthesis (DBT), etc.

This overview of breast lesion detection techniques compares them with standard mammography, and presents some benefits and limitations of the various diagnostic modalities, which can enhance diagnostic results.

The Authors can consider some suggestions for the revision.

The Authors could provide a table that will summarize the most important advantages and disadvantages, indications/contraindications, and sensitivity/specificity of the different diagnostic techniques for breast lesion detection. This would be useful for clinical practice.

A list of the abbreviations at the end would be helpful.

Comments on the Quality of English Language

Minor editing of English language required

Author Response

Thank you for the suggestion. In both the first and this part, there is no analyzed clinical case; hence, we did not find it appropriate to include tables with numerical values. However, we have taken the suggestion to add more specific comparisons between the various techniques to provide a more comprehensive and in-depth overview. At the end of the chapter, there is a summary of the pros and cons of the techniques and various comparisons among them.

Furthermore, the language throughout the entire article has been reviewed, sources have been improved, and a substantial portion of the article has been reworked. The first part of the article has been cited. The abstract and conclusions have been enhanced.

This article is the second part of a previously published article with the same structure. In the first part, a physical analysis of mammographic technique was addressed. This is a continuation where the remaining techniques commonly used in breast imaging are explored from a physical perspective. Comparisons between techniques have been added to provide a more comprehensive overview, both from a physical standpoint and a diagnostic objective viewpoint.

Reviewer 4 Report (Previous Reviewer 1)

Comments and Suggestions for Authors

After the manuscript has been revised, the quality of the structure and content has been improved, and it can be considered for acceptance and publication.

Comments on the Quality of English Language

Some statements need to be double-checked and some minor grammatical issues need to be fixed.

Round 2

Reviewer 1 Report (New Reviewer)

Comments and Suggestions for Authors

All the corrections are included in the paper. Hence, the paper may be accepted in the present form.

This manuscript is a resubmission of an earlier submission. The following is a list of the peer review reports and author responses from that submission.

Round 1

Reviewer 1 Report

Comments and Suggestions for Authors

This paper mainly reviews the application of radiomics in breast cancer. The title of the paper is mainly on "future development", but in the main text, a lot of space is spent on the background of the development of artificial intelligence, machine learning, and deep learning. It is suggested that an analysis of the application of radiomics in breast cancer image processing need been added, and a detailed outlook and suggestions for future technological development also is needed.

In addition, it is recommended to modify the abstract of the article to be more in line with the content of the article, and the expression of some sentences is not suitable for placing in the abstract.

Author Response

The article aims to explore how artificial intelligence approaches the medical and diagnostic field. The in-depth study on artificial intelligence goes into specifics to highlight the technological development and the differences between the different approaches, and then delves into radiomics.

Additions have been made to the text and the abstract has also been modified 

Reviewer 2 Report

Comments and Suggestions for Authors

 This paper studies Radiomics in breast imaging future development (part II). Some weaknesses should be addressed in this paper. And answer the main question of how machine learning techniques have developed into radionics, which aims to propose a personalized diagnosis and treatment to the patient. Therefore, I suggest the authors resubmit it after a major revision. My suggestions are as follows:

1.     Your abstract is too short and needs to mention more about your contributions

2.     Your introduction is too short, and you should add at least three paragraphs for more clarification

3.      Please outline the structure of your paper at the end of the introduction with more details.

4.     You add a literature review part after the introduction. You mentioned your references in all parts. Please consider most parts of your references in the literature review part.

5.      I strongly suggest that the paper be proofread and reread meticulously again, particularly regarding the spelling and grammatical mistakes.

6.      Flowchart is beneficial; it’s also important to outline the methodology behind this new approach. Please consider a flowchart of your suggested approach at the beginning of your paper.

7.     I suggest that you update section 3 so that the illustration used in the methodology section should be more readable.

8.     Add the methodology part before considering the artificial intelligence (AI) Section. And consider AI.      

9.     Please clarify the definitions for the equation in line 377.  What is the main strategy behind this equation? Please consider a number for this equation as well.

10.  Please clarify the novelty behind your strategy. Some explanations about before AI approaches such as machine learning and deep learning are insufficient.

11.  Please explain more about Table 1.

12.  Following the mathematical model is difficult due to a few notational mistakes.

13.  Discuss the study's limitations and future research suggestions.

14.  To improve your related works, remove unrelated references and consider the following AI papers in your literature review:

- Personalized breast cancer treatments using artificial intelligence in radiomics and pathomics. Journal of Medical Imaging and Radiation Sciences. 2019 Dec 1;50(4):S32-41.

- Multi-region radiomics for artificially intelligent diagnosis of breast cancer using multimodal ultrasound. Computers in Biology and Medicine. 2022 Oct 1;149:105920.

-  An integrated artificial intelligence model for efficiency assessment in pharmaceutical companies during the COVID-19 pandemic. Sustainable Operations and Computers. 2022 Jan 1;3:156-67.

- Artificial intelligence-based clinical decision support systems using advanced medical imaging and radiomics. Current Problems in Diagnostic Radiology. 2021 Mar 1;50(2):262-7.

- Radiomics and artificial intelligence analysis with textural metrics extracted by contrast-enhanced mammography in the breast lesions classification. Diagnostics. 2021 Apr 30;11(5):815.

- PET-Derived Radiomics and Artificial Intelligence in Breast Cancer: A Systematic Review. International Journal of Molecular Sciences. 2022 Nov 2;23(21):13409.

- Optimized wavelet-based satellite image de-noising with multi-population differential evolution-assisted harris hawks optimization algorithm. Ieee Access. 2020 Jul 17;8:133076-85.

- A novel hybrid parametric and non-parametric optimisation model for average technical efficiency assessment in public hospitals during and post-COVID-19 pandemic. Bioengineering. 2021 Dec 27;9(1):7.                                                   

- Artificial intelligence and radiomics in pediatric molecular imaging. Methods. 2021 Apr 1;188:37-43.

In conclusion, this version is unacceptable and needs to apply all the suggested comments point by point. In particular, applying the suggested high-quality related references

Author Response

1. the abstract has been modified

2.     The introduction has been supplemented 

3.      Structure has been outlined at the end of the introduction

4.     As  template the references are in order at the end of the article 

5.      Errors have been corrected 

6.      The flowchart has been integrated

7.   Section 3 has been updated 

8. the part on integrated methodology.      

9.    There is no equation on line 377, and the equations are all labelled

10.  We have expanded on the strategy in the initial section 

11.  We added the table explanation 

12. There is no error in the mathematical model, as  reference 

13.  We have added a discussion of limits and developments

14.  We have added some new references 

Round 2

Reviewer 1 Report

Comments and Suggestions for Authors

As a review article, although the manuscript has been modified to some extent, there is still a certain gap between the content of the manuscript and the title of the article. There are some minor grammar and typographical problems when revising the paper. I don't think the current manuscript is suitable for publication.

Reviewer 2 Report

Comments and Suggestions for Authors

Dear Editor,

I have not seen any improvements in this manuscript and my comments as well. However, they just added more than 30 self-citations by the following co-authors:

1. Vincenzo Cuccurullo (13 self-citations)

2. Gianluca Gatta ( 8 self-citations)

3. Graziella Di Grezia ( 8 self--citations)

They just provided weird explanations without highlighting anything and this is unacceptable for such an outstanding journal.

I strongly suggest the rejection of this manuscript.